# Fluoride Modification of Titanium Surfaces Enhance Complement Activation

**DOI:** 10.3390/ma13030684

**Published:** 2020-02-03

**Authors:** Maria H. Pham, Håvard J. Haugen, Janne E. Reseland

**Affiliations:** Department of Biomaterials, Institute of Clinical Dentistry, University of Oslo, 0317 Oslo, Norway

**Keywords:** biomaterials, titanium, hydrofluoric acid, dental implants, buffy coat, acute reaction, surface modification

## Abstract

Immediately after dental implant insertion, blood will be in direct contact and interact with the implant surface and activates inflammatory responses and complement cascades within seconds. The aim of the present study was to determine the ability of fluoride-modified titanium surfaces to activate complement cascades using the human buffy coat as model. The buffy coats were exposed to hydrofluoric acid-modified surfaces for a short time and its responses were compared to controls. Identification and quantification of complement cascade biomarkers were conducted using ELISA kits and multianalyte profiling using Luminex. A lower level of C3 at 30 min and increased levels of C4, MIP-4, CRP, and pigment epithelium-derived factor at 360 min were found on modified surfaces as compared to controls. We found no significant differences in the levels of C3a, C5a, C Factor H, α2M, ApoA1, ApoC3, ApoE, Prealbumin, α1AT, and SAP in modified surfaces in the buffy coats. We conclude that titanium surfaces treated with hydrofluoric acid modify the levels of specific biomarkers related to the complement cascade and angiogenesis and, thus, tissue growth, remodeling and repair, as this may play a role in the enhanced clinical performance of fluoride-modified Ti dental implants.

## 1. Introduction

More than 5 million dental implants are inserted yearly in the U.S alone, according to Grand View Research. Currently fluoride treated surface dental implant (Osseospeed®, Dentsply, York, PA, USA) is a commonly used implant system worldwide [1,2,3]. Fluoride-treated dental implants have shown clinically successful results with increased bone-to-implant contact and functional bone attachment [4,5,6,7,8]. It has also been well reported that fluoride improve the peri-implant tissue responses [9,10,11,12,13]. Despite these satisfactory clinical results, the identification of molecular mechanisms behind the success is still unidentified. We have previously studied the effect of fluoride treatment on human primary osteoblasts and human gingival fibroblasts [14,15], however, without evidently different cellular responses to fluoride and non-fluoride treated surface.

It has been suggested that fluoride on titanium surface has a stimulating effect on osteogenesis by strengthening the thrombogenisity and improving the innate coagulation system [16]. Fluoride modification of surfaces have shown to improve the osseointegration in early stage of healing after insertion of dental implants [4]. Bone is a hard tissue where the remodelling and repairing process is continuously involving resorption of old bone by osteoclasts, and the formation of new bone by osteoblasts [17]. Early stages of wound healing process around dental implant is complex. The long-term success of an implant depends on the early classical stages of wound repair [18] and the establishment of a soft-tissue barrier to protect the underlying osseous structures makes osseointegration possible, surrounding the implant body [19].

The interaction of plasma proteins with artificial surfaces may trigger coagulation pathways by contact activation [20]. The various components of the complement system might have important roles in diverse biologic processes, ranging from early haematopoiesis to skeletal and vascular development. These biologic processes are crucial to early development of vascularization, tissue regeneration, and cell differentiation. Complement activation is also involved in induction from whole blood of chemokines and growth factors [21]. The surface modification by hydrofluoric acid (HF) has been reported to influence cell growth and adhesion of surrounding soft tissue to the implant surface [15].

Immediately after dental implant insertion, blood will be in direct contact and interact with the implant surface. Surface properties of biomaterials is known to activate inflammatory responses and complement cascades within seconds [22]. The buffy coat contains mainly white blood cells and platelets [21]. We hypothesize that a HF modified surface activates components in the complement system differently that non-modified titanium surfaces. In this study, we used a buffy coat blood model to identify soluble active components of the complement systems. This method can reveal the initial immunological processes at the surgical site and thus determine the cellular responses on the titanium surface. The aim of the present study is to determine the ability to activate complement systems using the human buffy coat model on HF-modified titanium surfaces.

## 2. Materials and Methods

### 2.1. Titanium Coin Production

Coins of grade 2 titanium (Ti) with a diameter of 14.0 mm and a height of 1.0 mm were grit-blasted with TiO_2_ 180–220 µm grain powder as previously described [23]. All the coins went through five washing steps as described previously [23]. One half (n = 108) of the coins went through an additional surface modification procedure with 0.2 vol% HF for 120 sec giving a fluoride-modified TiO_2_ surface (TiF) [24] whereas 108 coins were left untreated (Ti). All coins were sterilized by gamma-irradiation at 25 kGy with a Co-60 source (Institutt for Energi teknikk, Kjeller, Norway). The coins were placed in cell culture plates, y- sterilization, and kept dry. The washed coins were enwrapped and kept sterile prior to analysis and cell incubation. The coins have been fully characterized by a profilometer, Atomic Force Microscopy (AFM), X-ray Photoelectron Spectroscopy (XPS), contact angle, and secondary ion mass spectrometry (SIMS) in previous studies [14,24]. The number of coins in each group was chosen based on previous studies [14,15,25]. 

### 2.2. Ethics

The Regional Ethic Committee at the University of Oslo approved the use of human buffy coat for basal experiments (2014/699), and the experiments were performed in accordance with guidelines from Medical and Health Research Ethics (REC). 

### 2.3. Buffy Coat Model

Human undiluted buffy coats were obtained from whole blood of two healthy anonymous donors (Blood bank, University Hospital of Oslo Ullevål, Norway). As part of the standard procedure for the preservation of whole blood or red blood cells the anticoagulatant citrate phosphate dextrose (CPD) (0.25 mM) was added by the vendor, Ullevål Blodbanken, Oslo University Hospital (OUS, Oslo, Norway). The buffy coats were pooled, and aliquots of 1000 µL were added onto each of the non- and surface-modified HF Ti coins in 24-well cell culture plates. The plates were incubated for 30, 120, and 360 min at 37 °C under continuous movement. Complement activation was stopped by adding ethylenediaminetetraacetic acid (EDTA) (10 mM final concentration), followed by 15 min centrifugation in 3000 rpm, as previously described [26]. Aliquots of buffy coat were stored at −80 °C prior to analysis.

### 2.4. Levels of Biomarkers in Buffy Coat

Multianalyte profiling was performed using the Luminex 200™ system (Luminex Corporation, Austin, TX, USA) and acquired fluorescence data were analysed by the 3.1 × PONENT software (Luminex). 

The levels of biomarkers, transport molecules and plasma proteins: α2 Macroglobulin (α2M), apolipoprotein AI (ApoA1), apolipoprotein CIII (ApoC3), apolipoprotein E (ApoE), prealbumin, α1 antritrypsin (α1AT), serum amyloid protein (SAP), C reactive protein (CRP), macrophage inflammatory protein 4 (MIP-4), pigment epithelium-derived factor (PEDF), as well as selected factors in the complement cascade: complement C3 (C3), complement factor H (CFH), and complement C4 (C4) were determined by Milliplex map human neurodegenerative disease magnetic bead panel 1 (HNDG1MAG-36K) and panel 2 (HNDG2MAG-36K)(Millipore, Billerica, MA, USA). All analyses were performed according to the manufacturers’ protocols. The cytokines/factors (α2M, ApoA1, ApoC3, ApoE, α1AT, SAP, CRP, MIP-4, PEDF, C3, CFH, and C4) were quantified in buffy coat on the TiF coins (n = 12) and compared to’ the levels of the controls, buffy coat on Ti coins (n = 12), for each individual time point (30, 120, and 360 min). This setup was repeated three times.

### 2.5. Levels of C3a and C5

The levels of complement 3a (C3a) and complement 5a (C5a) were quantified using enzyme-linked immunosorbent assay kits (ELISA) from Quidel (San Diego, CA, USA) and BD bioscience (San Diego, CA, USA), respectively. C3a and C5a in the buffy coat on the fluoride-modified treated samples TiF (n = 12) were measured and compared to controls Ti (n = 12) on each time points (30, 120, and 360 min). This set up was repeated three times.

### 2.6. Statistical Analysis

The statistical analyses were conducted using SigmaPlot (V 14.0 for Windows, Systat Software Inc, Chicago, IL, USA). Luminex and ELISA analyses of the effect of fluoride-modified surface (TiF) (n = 12) was compared to the respective controls (Ti) (n = 12) at each time points. Normality and equal variance tests were performed for the surfaces effect on all of the parameters at each time point tested. Even though the majority of the parameters passed the normality tests, we have chosen to present all datasets using box plots displaying the median (Q2) and the interquartile range (IQR). Pairwise comparisons between groups and controls were made using the analysis of t-tests and comparison procedures with Tukey tests and Wilcoxon signed rank tests. The significance levels were notified as followed in the figures and tables with probability of *p* < 0.05. Every experiment in this study, as in a previous study [14,15], was performed three times.

## 3. Results

### 3.1. Complement Biomarkers

A lower level of C3 was found after exposure to TiF surfaces compared to non-modified surfaces at earliest time point, 30 min (*p* < 0.05). The median levels of C3 were reduced in all time points tested, however, due to high standard deviation, statistical significances were only obtained compared to controls at 30 min (Figure 1).

No statistically significant differences in the C3a and C5a levels in buffy coat on coins with TiF surfaces compared to controls were found at any time points (Figure 2 and Figure 3, respectively).

Component C4 is among the biomarkers of the complement cascade, which is involved with many functions of the innate system. A reduction of C4 levels in both modified and non-modified surfaces were found from time point 30 min to 120 min, but a significantly higher level of C4 was found at 360 min in TiF surfaces compared to non-modified surfaces (*p* < 0.001) (Figure 4).

Fluoride-modified Ti surfaces significantly enhanced the level of CRP in buffy coat compared to non-modified at time point 360 min. (*p* < 0.01) (Figure 5).

### 3.2. Angiogenetic Markers

PEDF has different effects on different tissues and cell types [27], and activates the classical complement pathway by binding to the head domains of C1q. In this study the levels of PEDF follow the same pattern as C4; with a significantly increased level of PEDF in fluoride-modified surfaces compared to controls at the latest time point, 360 min (*p* < 0.001) (Figure 6).

The MIP-4 levels appeared to be reduced on the non-modified surfaces from time point 30–360 min, but this trend was not statistically significant. The levels of MIP-4 were significantly higher at time points 120 and 360 min on TiF surfaces as compared to non-modified surfaces (*p* < 0.01) (Figure 7).

Factor H is important regulator of alternative coagulation pathway by recognizing self-surfaces to ensure that these are non-activating in the host tissue and inhibiting the complement amplification [26]. In this study we did not find any significant differences between factor H on fluoride-modified surfaces compared to controls at any time points tested (Figure 8).

### 3.3. Remodeling Biomarkers

No significant differences were found in levels of biomarkers, transport molecules and plasma proteins such as α2M, ApoA1, ApoC3, ApoE, prealbumin, α1AT, and SAP, in buffy coat on fluoride-modified Ti surfaces compared to non-modified Ti surfaces at any time points tested (data not shown).

## 4. Discussion

Hemocompatibility is one of the major criteria for clinical success in blood-contacting biomaterials [16]. There are many haematological analysis models for analysing the haemocompatibility of biomaterials [20]. In this study, anticoagulated buffy coats with EDTA were used to find the mechanisms of the complement cascade on surface modified Ti coins. This is concentrated blood after centrifugation consisting white blood cells and platelets. Other studies have investigated the response to surfaces with an alternative model, e.g., with whole blood models or isolated cells, like macrophages [16,28,29]

Our previous in vitro studies indicated that HF treatment on Ti surfaces had no effect on osteoblasts [14], while fluoride modification enhanced the proliferation of human gingival fibroblasts [15] on Ti surfaces. Blood contact is an important part of reparation and regeneration at an implant site. In this study, we used buffy coat since it is one of many excellent models to investigate complement systems on biomaterial surfaces like Ti [30,31,32,33,34,35,36]. Complement systems and macrophage activities are closely related to inflammatory responses [37].

A summary of fluoride-modified Ti surface for enhanced availability of biomarkers for complement cascade in this study is shown in Figure 9.

In the current study, we found a significant difference in of the levels of C3 from buffy coat on HF non-modified surfaces compared to modified surfaces. We may speculate that HF treated surfaces have slower activation of C3 in the initial state. According to Liu et al., the deposition of C3 will determine the activation of the complement system [38]. C3 is a central protein of the complement system, which is important to immune defence and provides a link between innate and adaptive immunity. The complement cascade consists of three different pathways: the classical, lectin, and the alternative pathway. The classical and lectin pathway can be activated through antibody binding or by pattern recognition, while the alternative pathway can be activated by C3 directly [39]. C4 is also responsible for the cleavage of C3 [40]. A significantly higher release of complement biomarkers as C4, CRP, PEDF, and MIP-4 on modified surfaces were observed in buffy coat at 360 min compared to controls. The increased amount of complement biomarkers confirmed an activation of complement cascade at latest time point. Based on our results we propose that non-modified surfaces might have an initial effect on activation of the complement system, but the modified surfaces induced activation of later time point although no evidence exists on this. To answer this question, future studies, specifically aimed to answer this question, need to be performed.

Angiogenesis is crucial necessary for bone formation and regeneration in both normal and pathologic physiology. According to Hilborg and Bjursten, inflammation stimulates complement activation and, hence, the activation of angiogenesis [41]. Pigment epithelium-derived factor or plasminogen activator inhibitor (PEDF) is an inductor of endothelial cell apoptosis. This factor is also a part of anti-angiogenic factors to balance angiogenesis and interfere with excessive vessel formation [42]. In this study, we found increased secretion of PEDF for fluoride-modified surface at 360 min exposure (Figure 8). Increased angiogenesis is an important factor successful integration of implant in bone.

The complement system is easily activated upon contact with biomaterial surfaces. It can be triggered by both biomaterials and cell surfaces in contact with blood. Thromboinflammation is suggested to be reduced by specific inhibitors bound to surfaces and may present a challenge to the innate immune system [43]. The complement system and the coagulation system play central roles during the events of an inflammatory response. While the complement system confers immunoprotective and regulatory functions, the coagulation cascade is responsible to ensure haemostatic maintenance. These systems are important for reparation and regeneration on artificial surfaces [37]. A significantly increased release of CRP was found in buffy coats on fluoride-modified surfaces compared to non-modified surfaces at 360 min. The classic and the alternative lectin complement pathway can either be activated in an immune complex-dependent and -independent manner. CRP is a C- reactive protein or a surface- bound pentraxins from IgG and IgM. CRP is known to interact with the complement system [44,45,46]. Nilsson et al. found that biomaterial surface may induce inflammation triggered by CRP [40]. Another study showed that biomaterial implants induce inflammation marker CRP at the site of implantation [47]. Since CRP is an acute phase reactant, Goel et al. used CRP as an indicative tool for measuring infection post op and prediction the outcome of dental implants [48]. CRP can also measure the incidents of implant-associated infections in orthopaedic surgery with iodine-supported titanium implants [49].

Chemokines are a family of low molecular weight cytokines that regulate the migration and activation of leukocyte populations in physiological and pathological conditions. They are primarily responsible for chemo attraction of specific subsets of leukocytes to the sites of inflammation [50]. MIP-4 maturate cultured monocytes to macrophages [50], and are essential for effective tissue regeneration as they regulate the recruitment, proliferation and differentiation of target cells [51]. The function of the macrophage inflammatory protein 4 (CCL18) or MIP-4 is poorly understood on biomaterial surfaces, especially on titanium surfaces. Wimmer et al. suggested that MIP-4 activated macrophage may cause a limited inflammatory response and modify tissue remodelling in vivo [52]. In this current study, there were significantly higher levels of MIP-4 at 120 and 360 min in fluoride-modified Ti surfaces compared to controls. These findings were observed at the latest time points and could have been more profound with longer exposure times. 

No significant differences were found between modified and non-modified surfaces concerning the deposition of C3a and C5a levels in the buffy coats. Both anaphylatoxins C3a and C5a, pro-inflammatory polypeptides, are responsible for molecular mechanisms of activation and regulation of the complement system [39]. It has also been suggested that these anaphylatoxins regulate cytokine secretion for the complement system [53,54,55]. Significantly different levels of C3, but not for C3a and C5a, were found at any time points tested. This may reveal one of the weaknesses of this study, e.g., the detection and measurements of biomarkers. Weber et al. suggested that these components could be bound to the surfaces [20] and it could be an indirect cause to the lack of differences for C3a and C5a on the various surfaces. 

Enhanced biomarkers in complement cascade and angiogenesis when compared to non-modified titanium could be one of the reasons why one can see an enhanced clinical performance of fluoride-modified titanium dental implants, as complement cascade is imperative for regulation of the of the tissue remodelling and modulation, which will influence the osseointegration in the next steps. 

There were some clear limitations of this study. Statistical analysis and biological considerations are shown to be complementary rather than contradictory [56], but depends highly on the experimental design. Based on this in vitro set up we cannot conclude that this will be the events of importance in vivo. In this experiment the time points study were 30, 60, 120, and 360 min. This study model may exclude the earliest blood to material interaction. Timing of the measurement may be an important factor since there will be early interactions between blood protein and Ti surfaces [28,29].

## 5. Conclusions

Titanium surfaces treated with hydrofluoric acid active enhanced biomarkers in complement cascade and angiogenesis when compared to non-modified titanium.

## Figures and Tables

**Figure 1 materials-13-00684-f001:**
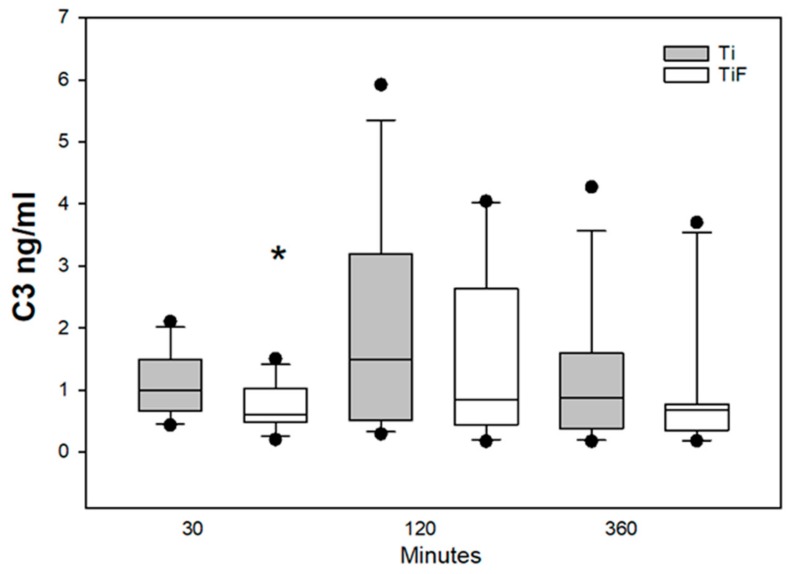
C3 levels in buffy coat on non-modified surfaces (Ti) (n = 12 per time point) and fluoride-modified surfaces (TiF) (n = 12 per time point). Values represent the median ± IQR (*; *p* < 0.05 compared to control).

**Figure 2 materials-13-00684-f002:**
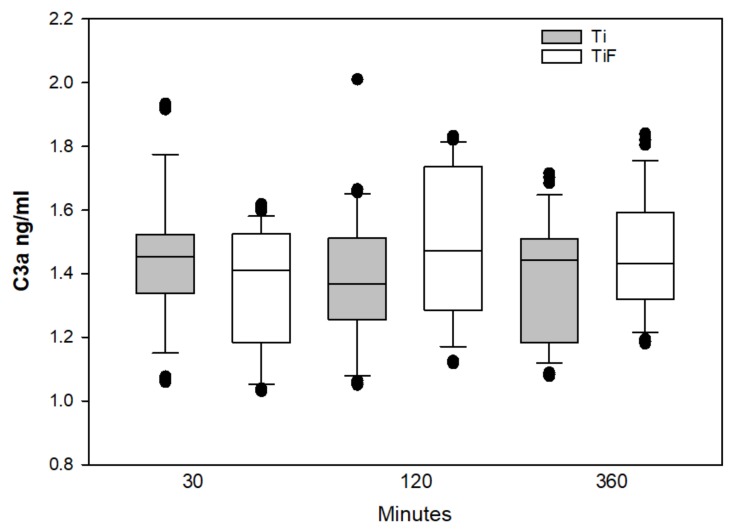
C3a levels in buffy coats on fluoride-modified (TiF) (n = 12 per time point) and non-modified (Ti) surfaces (n = 12 per time point). Values represent the median ± IQR (*p* < 0.05 compared to control).

**Figure 3 materials-13-00684-f003:**
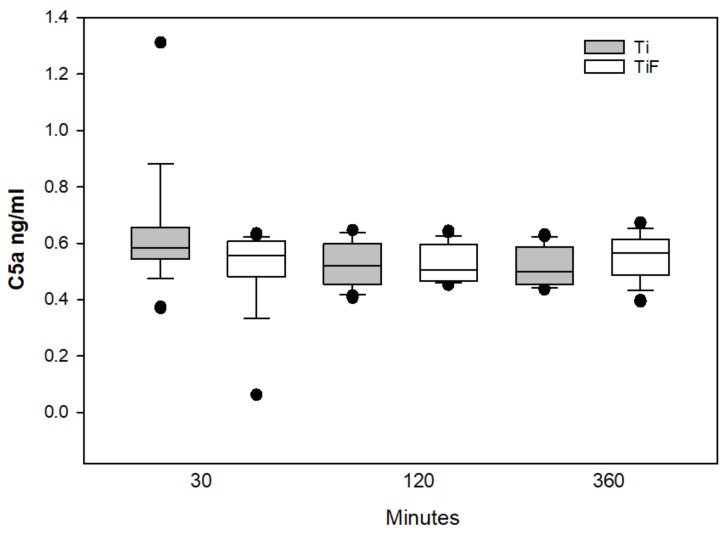
C5a levels in buffy coats on fluoride-modified (TiF) (n = 12 per time point) and non-modified (Ti) surfaces (n = 12 per time point). Values represent the median ± IQR (*p* < 0.05 compared to control).

**Figure 4 materials-13-00684-f004:**
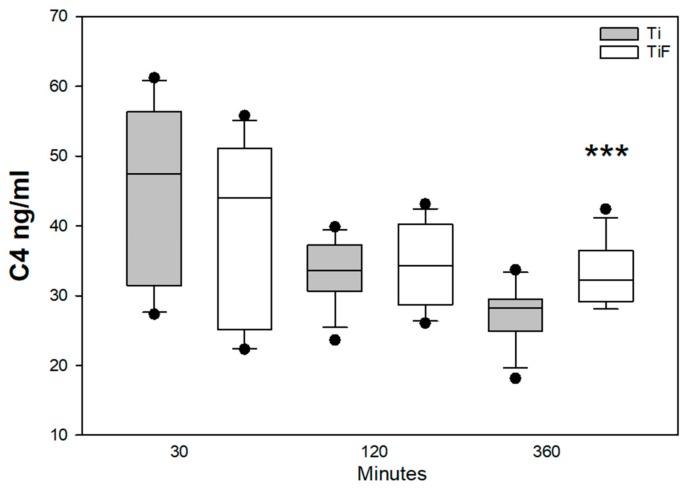
C4 levels in buffy coats on fluoride-modified (TiF) (n = 12 per time point) and non-modified (Ti) surfaces (n = 12 per time point). Values represent the median ± IQR (*** *p* < 0.001 compared to control).

**Figure 5 materials-13-00684-f005:**
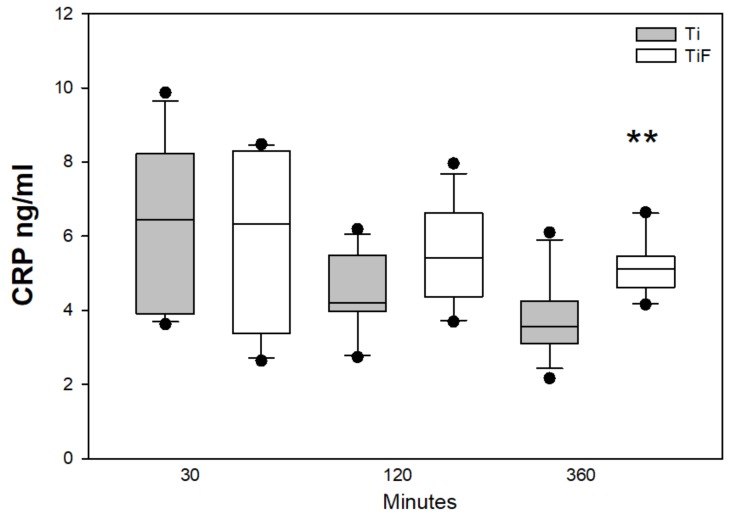
CRP levels in buffy coats on fluoride-modified (TiF) (n = 12 per time point) and non-modified (Ti) surfaces (n = 12 per time point). Values represent the median ± IQR (** *p* < 0.01 compared to control).

**Figure 6 materials-13-00684-f006:**
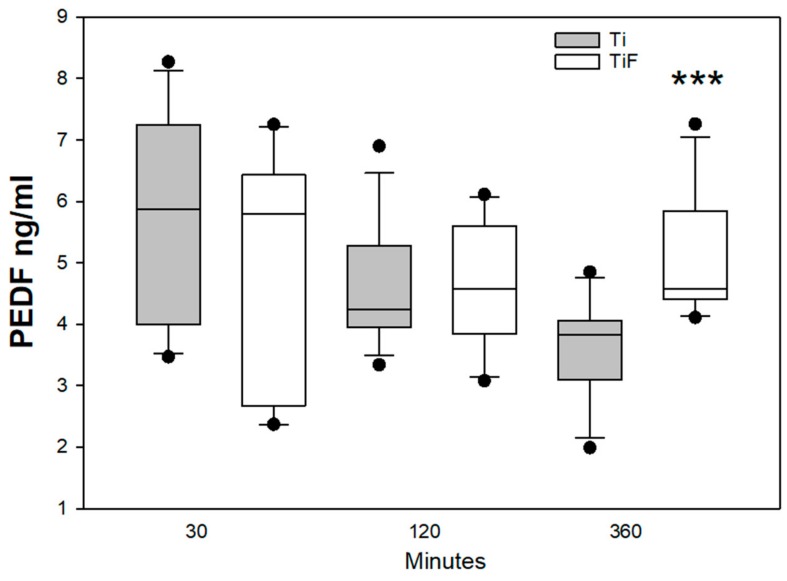
PEDF levels in buffy coats on fluoride-modified (TiF) (n = 12 per time point) and non-modified (Ti) surfaces (n = 12 per time point). Values represent the median ± IQR (***; *p* < 0.001 compared to control).

**Figure 7 materials-13-00684-f007:**
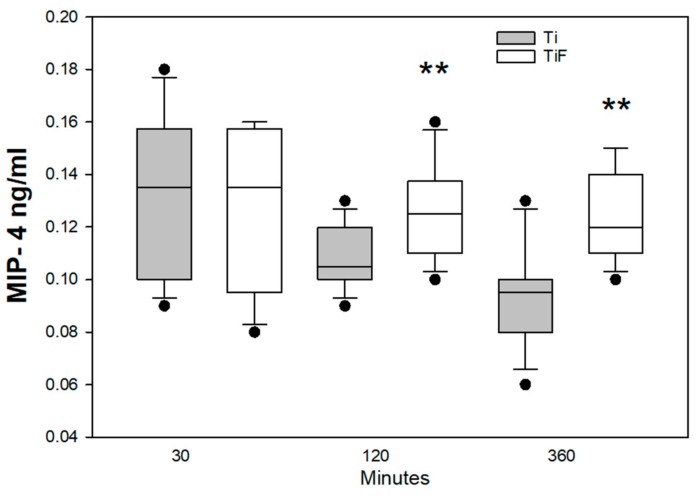
MIP-4 levels in buffy coats on fluoride-modified (TiF) (n = 12 per time point) and non-modified (Ti) surfaces (n = 12 per time point). Values represent the median. ± IQR (**; *p* < 0.01 compared to control).

**Figure 8 materials-13-00684-f008:**
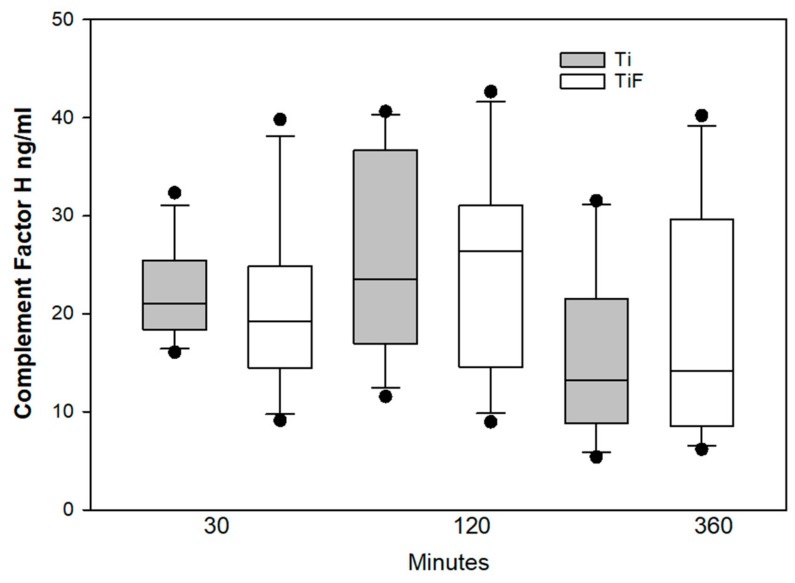
Complement factor H levels in buffy coats on fluoride-modified (TiF) (n = 12 per time point) and non-modified (Ti) surfaces (n = 12 per time point). Values represent the mean ± SD (*p* < 0.05 compared to control).

**Figure 9 materials-13-00684-f009:**
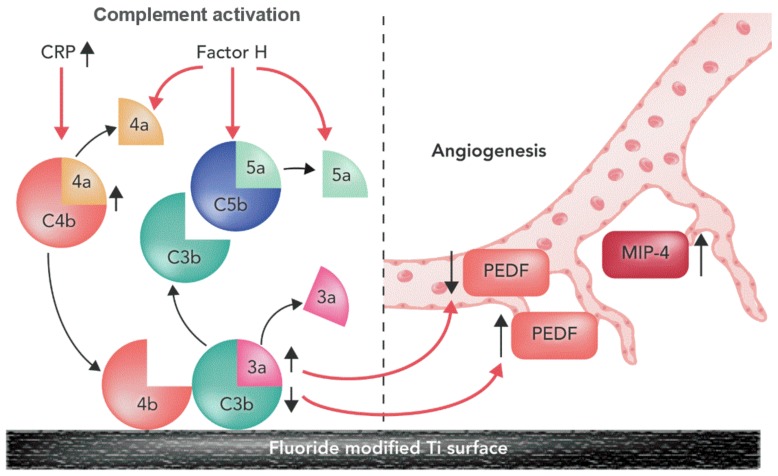
Fluoride-modified Ti surfaces increase availability of biomarkers for complement cascade and angiogenesis: C4, CRP, complement factor H, PEDF and MIP-4, but reduced levels of C3. PEDF is regulated by C3, C3a increases VEGF and decreases PEDF mRNA (as shown with black arrows). The biological factors, interaction between them, and their role (red arrows) are based on the current literature [37].

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
