# Peer review of "Fluoride Modification of Titanium Surfaces Enhance Complement Activation"

_materials, 2020, doi:10.3390/ma13030684_

Round 1

Reviewer 1 Report

The manuscript is very interesting as authors considered use of buffy coats, in order to consider angiogenesis and complement activation of flouride modified titanium surface. Despite well organized experiments, there are some concerns;

Introduction In Line 23, authors stated that “Currently fluoride treated surface dental implants are amongst the most popular implant 23 insertions worldwide[1].” This is not true. The reference is for just dental implant and I have never seen fluoride coated dental implant in clinic yet. Please review the reference again. In Line 45 to 47, it seems the information provided is completely different to Line 40 to 45. How complement activation linked to HF surface modification in terms of cell growth and adhesion? Despite all information in Introduction, what would be the novelty of this study? Perhaps stating hypotheses (null hypotheses) would be helpful. Materials and Methods Why authors considered Grade 2 Ti rather than Grade 4, which would be more relevant to currently used Ti for dental implant. Have authors considered using SLA surface treatment? How 1000 ul of buffy coats are added onto each of the Ti coins? Would there be space between coin and 24-well culture plate? Would HF coating influence wettability of the sample? The picture (photograph) of above experiment where 1000 ul of buffy coats are placed on Ti would be useful along with some contact angle testing. Results It may be feasible to have a summary table showing all complement system rather than having 8 figures for each complement. Figure 2, 3 and 8 figure legend, “p< 0.05 compared to control” can be deleted as there were no significant differences. Discussions The results seems very preliminary with one test to conclude what has been proposed to Figure 9. What would happened if I would have HF coating glass with same process? Would I have same results? Then HF coating has such effect. If not, Ti would have some play. Most of discussion is concerning text book information on importance of angiogenesis and complement system, and cannot see how the line 183 to 217 are related to this study. Discussion shall be reconsidered with the results obtained in this study only.

Author Response

Dear Editor

We are grateful for the constructive feedbacks to Manuscript ID: materials-699722.Thank you for your time, effort, and meticulous refereeing of the current manuscript. We have tried to our very best knowledge and effort to realise all the comments and suggestions from the reviewer. Specific comments to the revision is given below. All changes in the manuscript are done with track-changes.

We do hope that the editor find the manuscript materials-699722 worthy to be published in your journal and hope that our corrections are in according to expectations.

Sincerely,

The Authors

----

Reviewer 1

Introduction in Line 23, authors stated that “Currently fluoride treated surface dental implants are amongst the most popular implant 23 insertions worldwide [1].” This is not true. The reference is for just dental implant and I have never seen fluoride coated dental implant in clinic yet. Please review the reference again.

Answer: We do disagree with the reviewer statement, as Dentsply’s Osseospeed® surface is an example for fluoride surface modified dental implants. The Osseospeed® surface is produced identical on how the surfaces in this study has been made. The reason why we know this is that Osseospeed® was developed in our lab by our colleague Prof. Rølla and Prof. Ellingsen. Osseospeed® has been on the market since 2005, first marketed by AstraTech AB in Sweden, now by Dentsply.

Market reports clearly states that the most selling dental implant systems in the world are SLActive from Straumann AG (25%), followed by Dentium (20%), Danaher (19%) and fourth most is in fact Osseospeed® from Dentsply ( 12%), see page 28 on the link below for verification of sales number.

https://www.straumann.com/content/dam/media-center/group/en/documents/annual-report/2018/2018_Straumann_Annual_report.pdf

The citation we show is similar to Market reports like Decision Group, Frost Sullivan etc, however these reports are not publically open and thus we selected this citation instead.

We have added the brand name “Osseospeed®” in the manuscript to avoid any further confusion.

 In Line 45 to 47, it seems the information provided is completely different to Line 40 to 45.

Answer: We do not quite understand what the reviewer is referring as complement activation is a part of the coagulation pathways. Therefore, it is difficult to amend these lines as requested.

How complement activation linked to HF surface modification in terms of cell growth and adhesion? Despite all information in Introduction, what would be the novelty of this study? Perhaps stating hypotheses (null hypotheses) would be helpful.

Answer: One see an enhanced clinical performance of fluoride modified Ti dental implants. We wanted to look into if the complement cascade is imperative for regulation of the tissue remodeling and modulation.

Materials and Methods. Why authors considered Grade 2 Ti rather than Grade 4, which would be more relevant to currently used Ti for dental implant.

Answer: This is the same Ti grade as in Osseospeed® (Densply), see comments above regarding producing surfaces that are identical to Osseospeed®.

Have authors considered using SLA surface treatment?

Answer: We have cooperation with Straumann and do have access to some coins that have the SLActive surface. However, we have not been given permission to use those coins for these experiments nor do we have enough samples. The surface treatment behind SLActive is a protected surface treatment and since the method has niether been patented nor published, it is difficult to replicate this surface with 100% degree certainty. Since we know the details regarding the Osseopeed® surface, these were done instead.

We do hope that we in the future will able to do similar work for Straumann with their surfaces. Particular when we can show that we see differences for Osseospeed surfaces.

How 1000 ul of buffy coats are added onto each of the Ti coins?

Would there be space between coin and 24-well culture plate? Would HF coating influence wettability of the sample? The picture (photograph) of above experiment where 1000 ul of buffy coats are placed on Ti would be useful along with some contact angle testing.

Answer: The buffy coat were added drop-wise onto coins in 24-well cell culture plates. Profilometer, AFM, XPS, contact angle and SIMS analysis of these HF modified coins have been done in previous studies performed by us, and they were cited in the reference list (e.g. Lamolle et al). Information about these studies has been added into the revised manuscript.

Results. It may be feasible to have a summary table showing all complement system rather than having 8 figures for each complement. Figure 2, 3 and 8 figure legend, “p< 0.05 compared to control” can be deleted as there were no significant differences.

Answer:  Taken into consideration and corrected in the revised manuscript.

Discussions The results seems very preliminary with one test to conclude what has been proposed to Figure 9. What would happened if I would have HF coating glass with same process? Would I have same results? Then HF coating has such effect. If not, Ti would have some play.

Answer: The hypothesis in this study was to find molecular mechanisms why fluoride treated titanium surfaces (Osseospeed®) perform better than non-fluoride treated surfaces (TiOBlast® from AstraTech), as our previous study has not been able to single out any of those factors. As stated above, this is a frequently used surface in dental practices. The reason why glasses was not introduced was that such surfaces are not a surface that dental surgeons are using, and one can see by the cited market reports that the bioactive glass coated dental implants have a hardly any market share at all. Another point is that HF is very aggressive towards glasses as HF attacks the SiO bonds and hence such an experiment would be difficult to perform without disrupting the coating.

Most of discussion is concerning textbook information on importance of angiogenesis and complement system, and cannot see how the line 183 to 217 are related to this study. Discussion shall be reconsidered with the results obtained in this study only.

Answer: We disagree with this comment, as the intention with this section is to explain the link between the complement system and angiogenesis, and the role of the factors we have studied.

Angiogenesis is crucial for a successful osseointegrated dental implant and particular when one consider the long-term success rate of dental implant.  However, we do agree, as pointed out by reviewer 2 below that we have not showed or proved angiogenesis, but looked at biomarkers for angiogenesis. The title and the discussion has been amended to reflect this point. In due course, we have indeed showed effect on biomarkers for angiogenesis and this we do think it is compelling to discuss angiogenesis in the discussion. E.g., there is a paragraph on PEDF, which we also measured in our study.

Reviewer 2 Report

I suggest the title to be changed to better reflect the experimental work done. For example, the angiogenesis was not directly evaluated. What was evaluated were changes in some biomarkers related to angiogenesis. For example, something as: "Fluoride modification of titanium surfaces enhances the availability of biomarkers of complement activation or angiogenesis”

Line 23: please add a reference

Please insert a space before the citation’s numbers

Line 38: insert an “a” before soft-tissue barrier

I suggest the title 2.1 to be changed

Line 60: I suggest the word “elsewhere” to be changed for “previously” for example

How was the number of specimens (n=108 for treated and non-treated) calculated? Did the authors perform a sample calculation test previous to the experimental work?

The authors refer at lines 93-95 that: “Even though the majority of the bone parameters passed the normality tests, we have chosen to present all datasets using box plots displaying the median (Q2) and the Interquartile range (IQR)." However, figures 1, 4, 5, 6, 7 and 8 are presented as mean +- standard deviation, accordingly to the figure's quotes. Is this correct? What did the authors mean with "bone parameters"?

Please put “p” from statistical significance, “per” and “et al.” in italic form

Line 100-101: please add the number of ethical approval

Line 101: authors refer that “the experiment was performed in accordance with guidelines”. Which guidelines are the authors referring to?

I suggest the point 2.6 – ethics to be moved upward on the text, before point 2.2, were human buffy coats were used

Figure 1: the *, referring to statistical differences at 30 min should be moved to the TiF groups, since it refers to differences to the control group

Lines 107-108: the authors refer that at 30 min, p = 0.05. However, in figure 1 caption the p-value is referred to as p < 0.05. What is correct?

Line 108: the authors refer that “the median levels of C3 were reduced in all time points tested”. What does this mean? They were reduced compared to a reference value that is not referred to? Or the authors consider the values low since they are around 1-2 ng/ml?

Line 140: I do not agree with the expression “failed to be statistically significant”. Please change it.

What are the reference values of the biomarkers evaluated? I believe this is important since the observed differences can have a statistical difference, but this means a biological relevant difference? Please add this information to the discussion section. 

Text at line 165 is not visible in the manuscript

A reference (or more than one) should be added at figure 9 caption

Lines 181-182: the authors refer: "We suggest that non-modified surfaces might have an initial effect on activation of the complement system, but the modified surfaces induced activation of later timepoint.” Can the authors provide some explanation for this suggestion? Why the modified surfaces induced a later activation?

Lines 197-206: the authors refer that CRP induces inflammation at the implant site, besides other effects. Since the authors refer that "CRP is known to interact with the complement system", it must be more clearly discussed how the observed increase in CRP affects the complement system.

Lines 227-234: since this information is about the experimental model, I suggest this information be moved upward the manuscript, to the beginning of the discussion section, before the discussion of the results

Lines 237-240: this information should be moved to the discussion section since it not refers to the conclusions of this study

Please add a paragraph regarding the study limitations to the discussion section

Plagiarism software found some issues regarding published papers: "The effect of hydrofluoric acid treatment of titanium and titanium dioxide surface on primary human osteoblasts” and “Alginate microsphere compositions dictate different mechanisms of complement activation with consequences for cytokine release and leukocyte activation”. Please changes lines 31-32; 36-39; 175-176 for example, besides others

Author Response

Dear Editor

We are grateful for the constructive feedbacks to Manuscript ID: materials-699722.Thank you for your time, effort, and meticulous refereeing of the current manuscript. We have tried to our very best knowledge and effort to realise all the comments and suggestions from the reviewer. Specific comments to the revision is given below. All changes in the manuscript are done with track-changes.

We do hope that the editor find the manuscript materials-699722 worthy to be published in your journal and hope that our corrections are in according to expectations.

Sincerely,

The Authors

----

Reviewer 2

I suggest the title to be changed to better reflect the experimental work done. For example, the angiogenesis was not directly evaluated. What was evaluated were changes in some biomarkers related to angiogenesis. For example, something as: "Fluoride modification of titanium surfaces enhances the availability of biomarkers of complement activation or angiogenesis”

Answer: Yes, we agree with this comment, the title has been corrected in the manuscript.

Line 23: please add a reference

Answer: Taken into consideration and corrected in the revised manuscript.

Please insert a space before the citation’s numbers

Line 38: insert an “a” before soft-tissue barrier

Answer: Taken into consideration and corrected in the revised manuscript.

I suggest the title 2.1 to be changed

Answer: Taken into consideration and corrected in the revised manuscript.

Line 60: I suggest the word “elsewhere” to be changed for “previously” for example

Answer: Taken into consideration and corrected in the revised manuscript.

How was the number of specimens (n=108 for treated and non-treated) calculated? Did the authors perform a sample calculation test previous to the experimental work?

Answer: 216 was the total amount of Ti coins. Half (108) were treated with hydrogen fluoride acid and half did not undergo treatment (used as controls). Two different studies were done. 1)24 coins in both groups (Ti and TiF) were done with 3 biological replicas (72x2 =144).  2) 12x3 = 36 for each group Ti and TiF  (36x2= 72) . 144+72= 216. N= 24 and N=12 per group/time point

The amount of coins/samples were based on  previous studies. The references are listed below, and information included in the revised version of the manuscript.

Pham, M.H.; Landin, M.A.; Tiainen, H.; Reseland, J.E.; Ellingsen, J.E.; Haugen, H.J. The effect of hydrofluoric acid treatment of titanium and titanium dioxide surface on primary human osteoblasts. Clin Oral Implants Res 2014, 25, 385-394, doi:10.1111/clr.12150.

Pham, M.H.; Haugen, H.J.; Rinna, A.; Ellingsen, J.E.; Reseland, J.E. Hydrofluoric acid treatment of titanium surfaces enhances the proliferation of human gingival fibroblasts. Journal of tissue engineering 2019, 10, 2041731419828950, doi:10.1177/2041731419828950.

The authors refer at lines 93-95 that: “Even though the majority of the bone parameters passed the normality tests, we have chosen to present all datasets using box plots displaying the median (Q2) and the Interquartile range (IQR)." However, figures 1, 4, 5, 6, 7 and 8 are presented as mean +- standard deviation, accordingly to the figure's quotes. Is this correct? What did the authors mean with "bone parameters"?

Answer: Sorry, this is of course typing errors. The correct text is of course “parameters”. “Values represent the median. ± IQR” All the figure legends has been corrected. Obviously, the median=mean when the datasets are normal distributed.

Please put “p” from statistical significance, “per” and “et al.” in italic form

Answer: Taken into consideration and corrected in the revised manuscript.

Line 100-101: please add the number of ethical approval

Answer: Taken into consideration and corrected in the revised manuscript.

Line 101: authors refer that “the experiment was performed in accordance with guidelines”. Which guidelines are the authors referring to?

Answer: The Regional Ethic Committee at the University of Oslo approved the use of human buffy coat for basal experiments, and the experiment was performed in accordance with guidelines from Medical and Health Research Ethics (REC). Taken into consideration and corrected in the revised manuscript.

I suggest the point 2.6 – ethics to be moved upward on the text, before point 2.2, were human buffy coats were used

Answer: Taken into consideration and corrected in the revised manuscript.

Figure 1: the *, referring to statistical differences at 30 min should be moved to the TiF groups, since it refers to differences to the control group

Answer: Taken into consideration and corrected in the revised manuscript.

Lines 107-108: the authors refer that at 30 min, p = 0.05. However, in figure 1 caption the p-value is referred to as p < 0.05. What is correct?

Answer: p<0.05 is corrected and revised in the manuscript.

Line 108: the authors refer that “the median levels of C3 were reduced in all time points tested”. What does this mean? They were reduced compared to a reference value that is not referred to? Or the authors consider the values low since they are around 1-2 ng/ml?

Answer: They were low because the group of fluoride modified was compared with the controls

Line 140: I do not agree with the expression “failed to be statistically significant”. Please change it.

Answer: Taken into consideration and corrected in the revised manuscript.

What are the reference values of the biomarkers evaluated? I believe this is important since the observed differences can have a statistical difference, but this means a biological relevant difference? Please add this information to the discussion section.

Answer: It is not clear what the reviewer mean by ‘reference value’ related to the analyses performed in this in vitro study.  Reference values for a given test are normally based on the results that are seen in 95% of the healthy population and used by doctors to interpret a patient's test results.

We have presented the absolute values from the various test, and the lower limit of detection (LOD) or minimum detectable dose (MDD) of the biomarkers in this study are defined by the standards provided by the kit.

The reviewer brings up an important point and this information is added to the discussion part.  Statistical analysis and biological considerations are shown to be complementary rather than contradictory (Lovell et al 2013), but depends highly on the experimental.

Text at line 165 is not visible in the manuscript

Answer: Taken into consideration and corrected in the revised manuscript.

A reference (or more than one) should be added at figure 9 caption

Answer: We have changed the legend to clarify that black arrows display the changes in biomarkers. Taken into consideration and corrected in the revised manuscript.

Lines 181-182: the authors refer: "We suggest that non-modified surfaces might have an initial effect on activation of the complement system, but the modified surfaces induced activation of later time point.” Can the authors provide some explanation for this suggestion? Why the modified surfaces induced a later activation?

Answer: This would be pure speculation from our side as we have not solid evidence to make a statement on why. We are basing our next study to try to answer this question. Maybe the rate and quality of blood protein adsorption play a role?

Lines 197-206: the authors refer that CRP induces inflammation at the implant site, besides other effects. Since the authors refer that "CRP is known to interact with the complement system", it must be more clearly discussed how the observed increase in CRP affects the complement system.

Answer: Our statement are based on the current knowledge, please see line 270-283 and figure 9.

Lines 227-234: since this information is about the experimental model, I suggest this information be moved upward the manuscript, to the beginning of the discussion section, before the discussion of the results

Answer: Taken into consideration and corrected in the revised manuscript.

Lines 237-240: this information should be moved to the discussion section since it not refers to the conclusions of this study

Answer: Taken into consideration and corrected in the revised manuscript.

Please add a paragraph regarding the study limitations to the discussion section

Answer: Taken into consideration and corrected in the revised manuscript.

Plagiarism software found some issues regarding published papers: "The effect of hydrofluoric acid treatment of titanium and titanium dioxide surface on primary human osteoblasts” and “Alginate microsphere compositions dictate different mechanisms of complement activation with consequences for cytokine release and leukocyte activation”. Please changes lines 31-32; 36-39; 175-176 for example, besides others

Answer: Taken into consideration and corrected in the revised manuscript.

Reviewer 3 Report

In abstract, authors said that "We found no significant differences in the levels of C3a, C5a, C Factor H, α2M, ApoA1, ApoC3, ApoE, Prealbumin, α1AT, and SAP in modified surfaces in the buffy coats". Then, how you can conclude that titanium surface treated with hydrofluoric acid modify the levels of specific 17 biomarkers.

Please describe additional surface modification procedure.

Detailed incubation condition was missing.

What kinds of anticoagulation method were applied upon purchase?

There should be a name of vendor for this purchase.  

How many samples were used for measuring each cytokine?  

In the statistical analysis section, total number of samples was 72. However, original number of coins was 216. Where did the others go?

Author Response

Dear Editor

We are grateful for the constructive feedbacks to Manuscript ID: materials-699722.Thank you for your time, effort, and meticulous refereeing of the current manuscript. We have tried to our very best knowledge and effort to realise all the comments and suggestions from the reviewer. Specific comments to the revision is given below. All changes in the manuscript are done with track-changes.

We do hope that the editor find the manuscript materials-699722 worthy to be published in your journal and hope that our corrections are in according to expectations.

Sincerely,

The Authors

----

Reviewer 3

I suggest the point 2.6 – ethics to be moved upward on the text, before point 2.2, were human buffy coats were used

Answer: Taken into consideration and corrected in the revised manuscript.

Figure 1: the *, referring to statistical differences at 30 min should be moved to the TiF groups, since it refers to differences to the control group

Answer: Taken into consideration and corrected in the revised manuscript.

Lines 107-108: the authors refer that at 30 min, p = 0.05. However, in figure 1 caption the p-value is referred to as p < 0.05. What is correct?

Answer: p<0.05 is corrected and revised in the manuscript.

Line 108: the authors refer that “the median levels of C3 were reduced in all time points tested”. What does this mean? They were reduced compared to a reference value that is not referred to? Or the authors consider the values low since they are around 1-2 ng/ml?

Answer: They were low because the group of fluoride modified was compared with the controls

Line 140: I do not agree with the expression “failed to be statistically significant”. Please change it.

Answer: Taken into consideration and corrected in the revised manuscript.

What are the reference values of the biomarkers evaluated? I believe this is important since the observed differences can have a statistical difference, but this means a biological relevant difference? Please add this information to the discussion section.

Answer: It is not clear what the reviewer mean by ‘reference value’ related to the analyses performed in this in vitro study.  Reference values for a given test are normally based on the results that are seen in 95% of the healthy population and used by doctors to interpret a patient's test results.

We have presented the absolute values from the various test, and the lower limit of detection (LOD) or minimum detectable dose (MDD) of the biomarkers in this study are defined by the standards provided by the kit.

The reviewer brings up an important point and this information is added to the discussion part.  Statistical analysis and biological considerations are shown to be complementary rather than contradictory (Lovell et al 2013), but depends highly on the experimental.

Text at line 165 is not visible in the manuscript

Answer: Taken into consideration and corrected in the revised manuscript.

A reference (or more than one) should be added at figure 9 caption

Answer: We have changed the legend to clarify that black arrows display the changes in biomarkers. Taken into consideration and corrected in the revised manuscript.

Lines 181-182: the authors refer: "We suggest that non-modified surfaces might have an initial effect on activation of the complement system, but the modified surfaces induced activation of later time point.” Can the authors provide some explanation for this suggestion? Why the modified surfaces induced a later activation?

Answer: This would be pure speculation from our side as we have not solid evidence to make a statement on why. We are basing our next study to try to answer this question. Maybe the rate and quality of blood protein adsorption play a role?

Lines 197-206: the authors refer that CRP induces inflammation at the implant site, besides other effects. Since the authors refer that "CRP is known to interact with the complement system", it must be more clearly discussed how the observed increase in CRP affects the complement system.

Answer: Our statement are based on the current knowledge, please see line 270-283 and figure 9.

Lines 227-234: since this information is about the experimental model, I suggest this information be moved upward the manuscript, to the beginning of the discussion section, before the discussion of the results

Answer: Taken into consideration and corrected in the revised manuscript.

Lines 237-240: this information should be moved to the discussion section since it not refers to the conclusions of this study

Answer: Taken into consideration and corrected in the revised manuscript.

Please add a paragraph regarding the study limitations to the discussion section

Answer: Taken into consideration and corrected in the revised manuscript.

Plagiarism software found some issues regarding published papers: "The effect of hydrofluoric acid treatment of titanium and titanium dioxide surface on primary human osteoblasts” and “Alginate microsphere compositions dictate different mechanisms of complement activation with consequences for cytokine release and leukocyte activation”. Please changes lines 31-32; 36-39; 175-176 for example, besides others

Answer: Taken into consideration and corrected in the revised manuscript.

In abstract, authors said that "We found no significant differences in the levels of C3a, C5a, C Factor H, α2M, ApoA1, ApoC3, ApoE, Prealbumin, α1AT, and SAP in modified surfaces in the buffy coats". Then, how you can conclude that titanium surface treated with hydrofluoric acid modify the levels of specific 17 biomarkers.

Answer: We found the factors C3, C4, CRP, MIP-4 and PEDF to be significantly different, and we have based out conclusion on these. These are the factors we have screened, and we never claimed that we found a difference in all of these 17 factors.

The turnover of C3a and C3b is high and they have a high binding capacity to surfaces. These factors may influence the fact that these levels could not be measured properly in this study, and therefore we could find any significant difference this these.

Please describe additional surface modification procedure. Detailed incubation condition was missing.

Answer: The surface modification procedure and the incubation condition is now referred to Lamolle SF, Monjo M, Rubert M, et al. The effect of hydrofluoric acid treatment of titanium surface on nanostructural and chemical changes and the growth of MC3T3-E1 cells. Biomaterials 2009; 30: 736-742. DOI: DOI 10.1016/j.biomaterials.2008.10.052. Taken into consideration and corrected in the revised manuscript.

What kinds of anticoagulation method were applied upon purchase?

There should be a name of vendor for this purchase. 

Answer: The buffy coat was anticoagulated with acid citric monohydrate upon purchase from Ullevål, Oslo University Hospital (OUS). Taken into consideration and corrected in the revised manuscript.

How many samples were used for measuring each cytokine? 

In the statistical analysis section, total number of samples was 72. However, original number of coins was 216. Where did the others go?

Answer: 216 was the total amount of Ti coins. Half (108) were treated with hydrogen fluoride acid and half didn’t undergo treatment (used as controls). Two different studies were done. 1)24 coins in both groups (Ti and TiF) were done with 3 biological replicas (72x2 =144).  2) 12x3 = 36 for each group Ti and TiF (36x2= 72). 144+72= 216. N= 24 and N=12 per group/time point. Taken into consideration and corrected in the revised manuscript.

Round 2

Reviewer 1 Report

Thank you for the reply and modifications according to my previous comments.

Here are some minor points that would be helpful for readers.

I can see that there are marketed product of fluoride coated dental implant. My concern is the word 'the most popular implant insertions worldwide'. It is understandable that authors used market report from Staumann, which stated that the market share of Dentsply is currently 12%. However, this does not mean it is one of the most popular implant. Also, Dentsply produce may other implant as they acquired both Astra and Sirona, that would be part of the sales figures. Would there be any other clinical or scientific paper that also stated that flouride coated dental implant is popular? That would be a better reference. I still think that authors would need to highlight findings in Discussions. Fluoride coated Ti would suppress C3 but enhance C4, MIP-4, CRP, Pigment epithelium-derived factor level. But no effects on rest. Why? This study is focusing on difference between modified and non-modified surface, and it would be helpful to consider the differences and discuss the results between them. I am not sure if Figure 9 is helpful as it is not only relevant with C4, MIP-4, CRP, Pigment epithelium-derived factor, compare to the control, and the event would be same for other complement factors for just Ti (non-modified).

Author Response

I can see that there are marketed product of fluoride coated dental implant. My concern is the word 'the most popular implant insertions worldwide'. It is understandable that authors used market report from Staumann, which stated that the market share of Dentsply is currently 12%. However, this does not mean it is one of the most popular implant. Also, Dentsply produce may other implant as they acquired both Astra and Sirona, that would be part of the sales figures. Would there be any other clinical or scientific paper that also stated that flouride coated dental implant is popular? That would be a better reference.

Answer: You are right that after the merger between AstraTech, Dentsply and Sirona, there are several implant system in that consortium, and the market figures from the this new company contains several dental implant systems. The reference from Straumann Annual report does not differentiate between the different Dentsply Sirona systems.  Nevertheless, the recent market reports show that Osseospeed® is a common used implant system (see new references). We have altered the text to saying «commonly used» instead of «amongst the most popular». 

You are also asking for a clinical or scientific paper that states, however such papers are referring to market reports as a clinical or scientific paper do not assess the market penetration of different implant systems. Some clinical papers would claim that the different brand are popular, but these statements are referenced by market analysis reports. Thus, we do not think it is appropriate to use clinical or scientific papers as reference, but instead using the primary source, namely market reports, which have actually assessed the use of the different dental implant systems and above all. New market report references has been added which provide facts that indeed show that Osseospeed® is one of the five most selling dental implant systems in the world.

I still think that authors would need to highlight findings in Discussions. Fluoride coated Ti would suppress C3 but enhance C4, MIP-4, CRP, Pigment epithelium-derived factor level. But no effects on rest. Why?

Answer: As always in science, when one do experiments more questions arises, as we are able to answer.   It is difficult for us to speculate why this is happening, and it is our understanding that additional experiments needs to be performed to fully understand and interpret the impact of the data. As we observe the significant differences in biological markers at different time points, we try modestly to describe the results instead of speculating on each of the individual factor’s influence.

We have indicated several weaknesses in our study. Although we cannot quantify differences in the effect of the two surfaces on several markers, we cannot exclude that there are differences at earlier or later time points.

This study is focusing on difference between modified and non-modified surface, and it would be helpful to consider the differences and discuss the results between them. I am not sure if Figure 9 is helpful as it is not only relevant with C4, MIP-4, CRP, Pigment epithelium-derived factor, compare to the control, and the event would be same for other complement factors for just Ti (non-modified).

Answer: We have had several request from previous reviewers to make a figure, which explain and summaries the influences between the factor and biological significance of this. This figure is meant to summaries the effect on complimentary system from the current study.

Reviewer 2 Report

Please insert a space before the citation’s numbers

Line 23: since Osseospeed are not the only implants with fluoride-treated surfaces, why did the authors choose to refer to this specific brand? 

I suggest the title 2.1 to be changed to be more specific, as stated in the first revision round. The authors refer to a change, but the V2 of the manuscript remains the same title. I suggest for example "Titanium coin production"

Since the number of coins/samples were based on previous studies, as the authors stated, I suggest this information be added to the manuscript, to justify the number of specimens used.

Lines 75-76: “upon purchase” is repeated. Please remove one.

Please put “p” from statistical significance, “per” and “et al.” in italic form along with the text

Lines 106-107: what did authors mean with “Every experiment in this study were performed with 3 biological replicas”?

Why some experiments were performed with n=24 and others with n=12 if the authors refer that all experiments were done 3 times?

Please refer to the number of specimens consistently: N or n (at line 100 is written n and at figures captions is N for example)

Lines 196-198: the authors refer: "Based on our results we propose that non-modified surfaces might have an initial effect on activation of the complement system, but the modified surfaces induced activation of later timepoint.” In the previous revision round, I have asked the authors if they can provide a possible explanation for this. The authors refer to the response to reviewers that no evidence can support this statement. I think this information should be added to the manuscript, for example: "based on our results…later timepoint, although no evidence exists on this". I believe it's important for the readers to know that, at the present moment, this cannot be supported. The authors may even refer to future perspectives/future studies to answer this question, in the discussion section.

The authors used anticoagulated buffy coats with EDTA, however, several studies refer that EDTA interferes with complement activation (https://doi.org/10.1309/AJCPXPD7ZQXNTIAL; https://doi.org/10.1016/S1080-8914(97)80037-3; PMID: 10097633). This is a critical point. Did the authors think that the experimental model was well-chosen? If this may influence the obtained results? A discussion about this topic should be added to the discussion section.

Author Response

Please insert a space before the citation’s numbers

Answer: Taken into consideration and corrected in the revised manuscript.

Line 23: since Osseospeed are not the only implants with fluoride-treated surfaces, why did the authors choose to refer to this specific brand? 

Answer:  Osseospeed® was developed in our lab by our colleagues, Prof. Rølla and Prof. Ellingsen. The modifications done on the coins using HF in this study  is identical to how the Osseospeed® surface is modified. The PhD thesis of Ms Pham has been on investigating the effect of fluoride with various models, see reference list

I suggest the title 2.1 to be changed to be more specific, as stated in the first revision round. The authors refer to a change, but the V2 of the manuscript remains the same title. I suggest for example "Titanium coin production"

Answer: Taken into consideration and corrected in the revised manuscript

Since the number of coins/samples were based on previous studies, as the authors stated, I suggest this information be added to the manuscript, to justify the number of specimens used.

Answer: This information is now included in section 2.1.

Lines 75-76: “upon purchase” is repeated. Please remove one.

Answer: Taken into consideration and corrected in the revised manuscript

Please put “p” from statistical significance, “per” and “et al.” in italic form along with the text

Answer: The typing error is corrected in the revised version, and more information included as requested by reviewer 3.

Lines 106-107: what did authors mean with “Every experiment in this study were performed with 3 biological replicas”?

Answer:   It means that every experiment were performed 3 times. This information has been edited for clarification in section 2.4 and in the presentation of the results.

Why some experiments were performed with n=24 and others with n=12 if the authors refer that all experiments were done 3 times?

Answer: We are sorry for being unclear and the typing errors. The effect of fluoride modified surface (TiF) (n=12) was compared to the respective controls (Ti) (n=12) at each time points Each of these experiments were performed 3 times.

Please refer to the number of specimens consistently: N or n (at line 100 is written n and at figures captions is N for example)

Answer: Information on the number of replica (n) is now consequently included in the sentences instead as a separate statement.

Lines 196-198: the authors refer: "Based on our results we propose that non-modified surfaces might have an initial effect on activation of the complement system, but the modified surfaces induced activation of later timepoint.” In the previous revision round, I have asked the authors if they can provide a possible explanation for this. The authors refer to the response to reviewers that no evidence can support this statement. I think this information should be added to the manuscript, for example: "based on our results…later timepoint, although no evidence exists on this". I believe it's important for the readers to know that, at the present moment, this cannot be supported. The authors may even refer to future perspectives/future studies to answer this question, in the discussion section.

Answer: The manuscript has been amended accordingly

The authors used anticoagulated buffy coats with EDTA, however, several studies refer that EDTA interferes with complement activation (https://doi.org/10.1309/AJCPXPD7ZQXNTIAL; https://doi.org/10.1016/S1080-8914(97)80037-3; PMID: 10097633). This is a critical point. Did the authors think that the experimental model was well-chosen? If this may influence the obtained results? A discussion about this topic should be added to the discussion section.

Answer: We did not use used anticoagulated buffy coats with EDTA in the experiment. EDTA was added to stop the reactions at the time points of harvest. This is the same procedure as described by others - ref 26

Reviewer 3 Report

Some concerns raised by previous review had been ignored. Proper answers should be shown in all raised questions.

I attached your reply to my critics. You can find what was missing.

Please describe additional surface modification procedure. Detailed incubation condition was missing.

Answer: The surface modification procedure and the incubation condition is now referred to Lamolle SF, Monjo M, Rubert M, et al. The effect of hydrofluoric acid treatment of titanium surface on nanostructural and chemical changes and the growth of MC3T3-E1 cells. Biomaterials 2009; 30: 736-742. DOI: DOI 10.1016/j.biomaterials.2008.10.052. Taken into consideration and corrected in the revised manuscript.

What kinds of anticoagulation method were applied upon purchase?

There should be a name of vendor for this purchase. 

Answer: The buffy coat was anticoagulated with acid citric monohydrate upon purchase from Ullevål, Oslo University Hospital (OUS). Taken into consideration and corrected in the revised manuscript.

How many samples were used for measuring each cytokine? 

In the statistical analysis section, total number of samples was 72. However, original number of coins was 216. Where did the others go?

Answer: 216 was the total amount of Ti coins. Half (108) were treated with hydrogen fluoride acid and half didn’t undergo treatment (used as controls). Two different studies were done. 1)24 coins in both groups (Ti and TiF) were done with 3 biological replicas (72x2 =144).  2) 12x3 = 36 for each group Ti and TiF (36x2= 72). 144+72= 216. N= 24 and N=12 per group/time point. Taken into consideration and corrected in the revised manuscript.

Author Response

Some concerns raised by previous review had been ignored. Proper answers should be shown in all raised questions.

I attached your reply to my critics. You can find what was missing.

Please describe additional surface modification procedure. Detailed incubation condition was missing.

Answer: This information is now included in the method part.

What kinds of anticoagulation method were applied upon purchase?

Answer: The whole blood was anticoagulated with citrate phosphate dextrose (CPD) (0.25 mM) by the vendor, Ullevål Blodbanken, Oslo University Hospital (OUS). This information is now included.

This is the standard procedure for preservation of whole blood or red blood cells for up to 21 days. Buffy coats are isolated from whole blood at the blood bank by an initial centrifugation without a density gradient, and will consequently contain the same amount of anticoagulant.

There should be a name of vendor for this purchase. 

Answer: This is information that is not available for us. We do include a picture of the s buffy coat we got from our supplier.

These plastic bags from Macopharm have the anticoagulant included, but their supplier of this product is not identified.

How many samples were used for measuring each cytokine? 

Answer: Please see answer to reviewer 1. This information is now elaborated in the text

In the statistical analysis section, total number of samples was 72. However, original number of coins was 216. Where did the others go?

Answer: Answer: 72 coins were used for each of the 3 repetitions of the experiment (72 x 3=216).

We are sorry for being unclear and for the typing calculation error in the previous answer and  the manuscript.
